# Application of the skills network approach to measure physician competence in shared decision making based on self-assessment

**Levente Kriston** \*, **Lea Schumacher**, **Pola Hahlweg, Martin Härter, Isabelle Scholl**

Department of Medical Psychology, University Medical Center Hamburg-Eppendorf, Hamburg, Germany

☯ These authors contributed equally to this work.

* l.kriston@uke.de

**Data Availability Statement:** All data and the analysis code are available at https://osf.io/z7368/files/osfstorage.

**Funding:** The original study, of which data were used for analysis, was funded by the German

## Abstract

Several approaches to and definitions of 'shared decision making' (SDM) exist, which makes measurement challenging. Recently, a skills network approach was proposed, which conceptualizes SDM competence as an organized network of interacting SDM skills. With this approach, it was possible to accurately predict observer-rated SDM competence of physicians from the patients' assessments of the physician's SDM skills. The aim of this study was to assess whether using the skills network approach allows to predict observer-rated SDM competence of physicians from their self-reported SDM skills. We conducted a secondary data analysis of an observational study, in which outpatient care physicians rated their use of SDM skills with the physician version of the 9-item Shared Decision Making Questionnaire (SDM-Q-Doc) during consultations with chronically ill adult patients. Based on the estimated association of each skill with all other skills, an SDM skills network for each physician was constructed. Network parameters were used to predict observer-rated SDM competence, which was determined from audio-recorded consultations using three widely used measures (OPTION-12, OPTION-5, Four Habits Coding Scheme). In our study, 28 physicians rated consultations with 308 patients. The skill 'deliberating the decision' was central in the population skills network averaged across physicians. The correlation between parameters of the skills networks and observer-rated competence ranged from 0.65 to 0.82 across analyses. The use and connectedness of the skill 'eliciting treatment preference of the patient' showed the strongest unique association with observer-rated competence. Thus, we found evidence that processing SDM skill ratings from the physicians' perspective according to the skills network approach offers new theoretically and empirically grounded opportunities for the assessment of SDM competence. A feasible and robust measurement of SDM competence is essential for research on SDM and can be applied for evaluating SDM competence during medical education, for training evaluation, and for quality management purposes. [A plain language summary of the study is available at https://osf.io/3wy4v.]

Federal Ministry of Education and Research (https://www.bmbf.de/bmbf/en/home/home_node.html, grant number: 01GX0742, grant received by LK, MH and IS). The secondary analysis presented here was not externally funded. The funder of the original study had no role in study design, data collection and analysis, decision to publish, or preparation of the manuscript.

**Competing interests:** I have read the journal's policy and the authors of this manuscript have the following competing interests: LK, MH, and IS report academic, but not financial, conflict of interest as the developers of the investigated measure, the SDM-Q-Doc. This does not alter our adherence to PLOS ONE policies on sharing data and materials

## Introduction

An essential patient-centered communication competence in health care delivery is the ability to support shared decision making (SDM) in medical consultations. SDM is frequently described as an interpersonal decision making process with a strong emphasis on a balanced flow and exchange of information, values, preferences, power, and responsibility between the patient and the health care professional during medical consultations [1, 2]. SDM has been considered ethical in medical consultations because it ensures that patients are informed about various treatment options and that the patients' preferences are valued in medical decision-making [3]. This seems particularly important considering that physicians´ assumptions on their patients' preferences often mismatch the patients' actual preferences, and patients tend to choose different treatment options when they are better informed [4]. Further, SDM may help to reduce the use of inappropriate tests and interventions when benefits and drawbacks of these are clearly discussed [5] and could lead to better medication adherence [6]. Finally, as patients tend to choose more conservative options when asked, SDM might even reduce health care costs [7]. Thus, it is not surprising that major health care organizations have adopted the principles of SDM [8–10].

Although several definitions of SDM exist, it is rarely acknowledged explicitly that the same term can refer to ontologically very different concepts [11]. It is frequently unclear, whether 'SDM' is used to denote observable attributes of the communication process in a medical encounter, the perception of these attributes by the patient or the physician, attitudes of the participating individuals, a specific method or technique which physicians can utilize, a general philosophy of shaping health care, or a scientific model of medical communication. Conceptual clarity is indispensable for the measurement of latent constructs [12]. From a competence-focused perspective, SDM competence can be defined as the physician's ability to use specific behavioral skills in a way which supports building a consensus with the patient regarding the favored treatment among multiple viable options in accordance with the patient's preferences and values [13]. According to this approach, SDM competence requires physicians to organize a defined set of behavioral skills into a certain pattern or network to make a patient-centered decision in the medical consultation more likely.

In a recent study, we found that modelling SDM competence as a network of skills can be used to predict physicians' observer-rated SDM competence [13]. In that study, patients rated the degree to which certain SDM-related skills were shown by the physicians in their routine medical consultations. These ratings were used to create an SDM skills network for each physician, which models how individual SDM skills are related to each other. Attributes of these networks, e.g., how strongly a skill was related to other skills, predicted observer-rated competence with high accuracy. Using this approach with skill rating input from other sources than patients would substantiate the validity of conceptualizing SDM competence as an organized network of behavioral skills. In the present study, we investigated whether processing physician-reported data on their SDM skills according to the skills network approach can be used to predict observer-rated SDM competence.

## Materials and methods

### Design and procedures

The design of the present study was based on a previous investigation [13]. We re-analyzed data from a study on measuring SDM, collected between August 2009 and September 2010 in Hamburg, Germany [14]. In that study, consultations between adult patients with chronic conditions who faced a treatment decision and physicians providing primary and specialty

outpatient care were examined using ratings from patients, physicians, and external observers. The investigators aimed to include thirty physicians with written documentation of ten consultations and audio-recordings of three consultations each. The ethics committee of the state chamber of physicians in Hamburg approved the study protocol (record no. PV3180). All participants provided written informed consent. In the present analysis, we used data from the physicians and the external observers.

## Measures

Basic demographic and clinical data on the participating patients and physicians were collected by administering written questionnaires.

Physician-reported data on SDM skills were collected with the physician version of the 9-item Shared Decision Making Questionnaire (SDM-Q-Doc), which was filled out after the respective consultations [15]. This measure requires physicians to rate the degree to which they showed nine behaviors in the consultation using a six-step Likert-type scale ranging from zero to five. The behaviors captured by the SDM-Q-Doc correspond to key SDM skills: focusing the decision, sharing the decision, presenting options, informing on options, supporting comprehension, eliciting preferences, deliberating the decision, selecting an option, and planning actions [13, 15].

Observer-rated SDM competence of the physicians was measured by three widely used validated measures, the OPTION-12 [16, 17], the OPTION-5 [18, 19], and the Invest in the End subscale of the Four Habits Coding Scheme (4HCS) [20, 21], based on the audio-recorded consultations. We decided to include all three measures in the present analysis, because they capture SDM competence from different perspectives. The OPTION measures focus on decision making, while the 4HCS assesses primarily communication. We decided to include the OPTION-5 in addition to the OPTION-12, because it has a stronger focus on patient preferences and is based on a revised model of SDM [18]. As shown also by empirical analysis, the OPTION-12, the OPTION-5, and the Invest in the End subscale of the 4HCS capture overlapping but notably distinct constructs [13].

Two independent raters assessed each consultation using pilot sessions and manuals to achieve sufficient agreement. Inter-rater reliability varied between 0.69 to 0.76 across instruments for averaged ratings of the physicians' SDM competence, showing substantial agreement between raters [13]. Raters were blinded to the results of the assessment with other measures. For analysis, we transformed all scores to range from 0 to 100, with higher values indicating a higher level of SDM competence. Each of the measures was averaged across consultations in order to obtain three observer-rated SDM competence scores for each physician. Validity of this method of estimating competence was supported by substantial physician-level variance of the three scores and moderate to high physician-level correlation between them [13]. This means that while SDM competence considerably varied between different physicians, the three observer-rated measures indicated similar SDM competence for each individual physician. An overview on the design, measures and analysis is displayed in Fig 1.

## Statistical analysis

The physicians' self-rated data on their SDM skills were analyzed according to the skills network model of competence [13]. We assessed the associations between the nine SDM skills and constructed a skills network for each physician. These networks display individual SDM skills as nodes. The connections between nodes are called edges, which indicate how strongly individual SDM skills are related to each other.

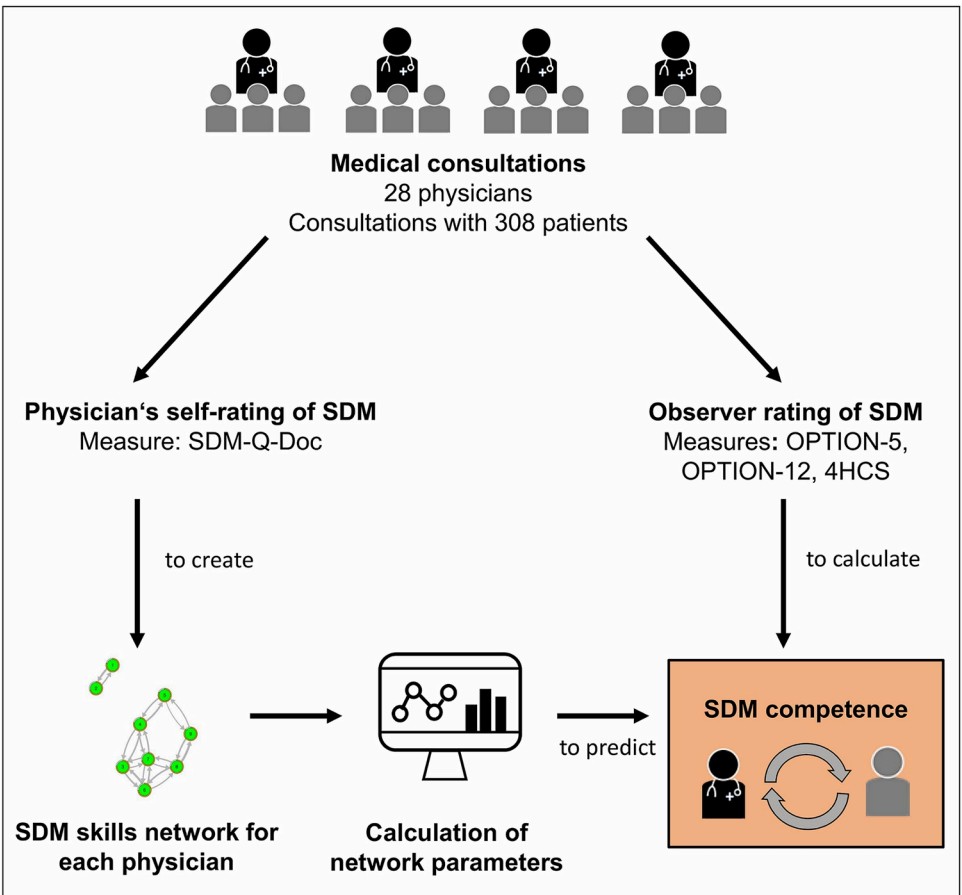

**Fig 1. Overview of the research design, measures and analysis.** SDM, Shared Decision Making; SDM-Q-Doc, Shared Decision Making Questionnaire—physician version; 4HCS, Invest in the End subscale of the Four Habits Coding Scheme.

For each SDM skill, a Bayesian multilevel linear regression was estimated with the skill as the outcome variable and all other skills as predictors, based on the physician-rated data from all consultations of all physicians. The intercept and slopes were allowed to vary between physicians yielding estimates for each individual physician. Thus, the strength of the associations between individual SDM skills was expected to vary across physicians. Bayesian analysis requires the definition of a prior distribution for each estimated parameter. This prior distribution is updated during the analysis by combining it with the observed data to obtain a posterior distribution, which informs on how probable certain values of the estimated parameter are. We used weakly informative priors, reflecting that we had an approximate but not exact idea of the expected size of the statistical parameters before calculation (see S1 File). If more than two of the nine skills were missing for a consultation, data points from that consultation were excluded. One or two missing ratings per consultations were imputed using the expectation-maximization algorithm.

Based on the estimated coefficients from the multilevel regression models described above, a skills network was constructed for each physician. The regression estimates describing the direction and strength of the association between the different skills for each physician were used as edge weights. When the 95% credible interval of a regression estimate included a zero, this association was excluded to avoid spurious associations. Nodes in the skills networks were placed using the Fruchterman-Reingold algorithm, thus, as far as possible in two dimensions,

their distance is relative to the strength of their association [22]. Consequently, skills that were strongly related were placed closer to each other in the networks. Three network parameters, namely activation, outstrength and instrength, of each skill for each physician were calculated. Activation of a skill was defined as the mean of that skill across consultations, i.e., how strongly each physician indicated to have used the skill across their consultations. Outstrength of a skill was calculated by summing the weights of the outgoing edges of that skill and indicates how strongly a skill influences other skills. Instrength was calculated by summing the weights of the ingoing edges of that skill, showing how strongly that skill is influenced by other skills. In addition to the physician-specific networks, we created a population network through averaging the network parameters across physicians. Thus, in addition to constructing a network for each physician, a population network showing how skills are related on average across all physicians was also created. A more detailed description including a step-by-step instruction for calculations can be found elsewhere [13].

Finally, we performed Bayesian linear regression analyses to test whether the network parameters of each physician can predict observer-rated SDM competence as measured with the OPTION-5, OPTION-12 and the Invest in the End subscale of the 4HCS. By doing so, we tested whether characteristics of the skills networks predicted the SDM competence of individual physicians as rated by external observers. First, a confirmatory model with the activation, outstrength and instrength of the skills 'focusing the decision', 'eliciting preferences' and 'deliberating the decision' as predictors was tested, since these skills were relevant in the previous analysis with patient-rated data [13]. We used informative priors with means and standard deviations estimated from the posterior distribution of the estimates observed in the analysis of the patient-reported data (see S1 File) [13]. Subsequently, we created an exploratory model to investigate whether ignoring previous results changes the conclusions substantively. For this, three Bayesian linear regression models were fitted for predicting each observer-rated measure of SDM competence with the activation, the instrength, and the outstrength of all skills as predictors, respectively. The network parameters of the skills, which were significant predictors in this first step for at least one of the observer-rated measures, were regressed onto the three observer-rated measures in the final exploratory model. Weakly informative priors were chosen for all exploratory analyses (see S1 File).

All analyses were conducted in *R* version 4.0.4 [23]. Bayesian (multilevel) regression analyses were conducted with the package *brms* utilizing Markov chain Monte Carlo sampling methods [24]. Networks were plotted using *qgraph* [25]. All regression models were run with four chains, a total of 20,000 iterations, a thinning rate of 10, and 12,000 burn-in simulations, resulting in a posterior sample of 2,000. For each model, the Gelman-Rubin potential scale reduction statistic [26] and traceplots were checked for convergence. We labeled a regression coefficient as statically significant when its 95% credible interval did not include zero. The *R* code of all analyses is available at https://osf.io/z7368/.

## Results

### Sample

In the original study, 33 physicians agreed to participate [14], of which 28 provided self-assessment of their SDM skills in 326 consultations. Ratings of 18 consultations were excluded as they had more than two missing data points, resulting in data from 308 consultations included in the analyses (on average 11 consultations per physician). Audio recordings were available from 24 physicians and 80 consultations (on average 3.3 consultations per physician).

Over 70 percent of the participating physicians (42.9 percent female, mean age 50.4 years) were specialized in family or internal medicine and less than one in four had 20 years or more

experience (Table 1). The majority of the patients in the investigated consultations (60.3 percent female, mean age 54.2 years) were married, had a low to medium formal education, and were employed or retired (Table 2). About one third of the patients were diagnosed with type 2 diabetes, chronic back pain, and depressive disorder, respectively. The subsample of the physicians and patients contributing audio-recorded consultations were comparable to the total sample.

## Population network of SDM skills

The average skills network (Fig 2) showed that the skills 'focusing the decision' and 'sharing the decision' were, despite their strong reciprocal association, disconnected from the remaining network, suggesting that these skills were only related to each other. 'Presenting options', 'informing on options', 'eliciting preferences' and 'selecting an option' were strongly connected, with 'deliberating the decision' being in the center of this skill cluster, showing a high level of interrelatedness between these skills. The skills 'supporting comprehension' and 'planning actions' were more peripheral in the skills network, as they were only related to 'informing on options' and 'selecting an option', respectively.

On average, the skill 'planning actions' were most frequently shown (Fig 3, panel A). 'Presenting options' had the strongest influence on other skills (Fig 3, panel B), and ´informing on options' was most strongly influenced by other skills (Fig 4, panel C). There was considerable variation between the physicians in their network structure and network parameters (Fig 3; skills networks of individual physicians can be seen in S1 Fig). Thus, how skills were related to each other differed between physicians.

## Confirmatory prediction of observed SDM competence from skills networks

The skill 'eliciting preference' played an important role in the prediction of observer-rated SDM competence in the confirmatory model, as its activation was significantly positively

**Table 1. Characteristics of participating physicians.**

|  | Total sample (n = 28) | | Sample providing audio recording (n = 24) | |
|---|---|---|---|---|
|  | n | (per cent) | n | (per cent) |
| *Sex* |  |  |  |  |
| female | 12/28 | (42.9) | 11/24 | (45.8) |
| male | 16/28 | (57.1) | 13/24 | (54.2) |
| *Age* |  |  |  |  |
| 30 to 39 years | 2/28 | (7.1) | 2/24 | (8.3) |
| 40 to 49 years | 13/28 | (46.4) | 13/24 | (54.2) |
| 50 to 59 years | 6/28 | (21.4) | 4/24 | (16.7) |
| > 60 years | 7/28 | (25.0) | 5/24 | (20.8) |
| *Specialty* |  |  |  |  |
| family medicine | 11/28 | (39.3) | 11/24 | (45.8) |
| internal medicine | 9/28 | (32.1) | 8/24 | (33.3) |
| orthopedics | 4/28 | (14.3) | 3/24 | (12.5) |
| psychiatry | 4/28 | (14.3) | 2/24 | (8.3) |
| *Experience* |  |  |  |  |
| < 10 years | 11/28 | (39.3) | 11/24 | (45.8) |
| 10 to 19 years | 11/28 | (39.3) | 9/24 | (37.5) |
| 20 to 29 years | 4/28 | (14.3) | 2/24 | (8.3) |
| 30 to 39 years | 2/28 | (7.1) | 2/24 | (8.3) |

**Table 2. Characteristics of participating patients.**

| | Total sample (n = 308) | | Sample providing audio recording (n = 80) | |
|---|---|---|---|---|
| | n[a] | (per cent) | n[b] | (per cent) |
| *Sex* | | | | |
| female | 184/305 | (60.3) | 50/78 | (64.1) |
| male | 121/305 | (39.7) | 28/78 | (35.9) |
| *Age* | | | | |
| < 20 years | 3/308 | (1.0) | - | - |
| 20 to 29 years | 13/308 | (4.2) | 3/78 | (3.8) |
| 30 to 39 years | 40/308 | (13.0) | 10/78 | (12.8) |
| 40 to 49 years | 61/308 | (19.8) | 16/78 | (20.5) |
| 50 to 59 years | 66/308 | (21.4) | 16/78 | (20.5) |
| 60 to 69 years | 65/308 | (21.1) | 20/78 | (25.6) |
| 70 to 79 years | 48/308 | (15.6) | 11/78 | (14.1) |
| > 79 years | 12/308 | (3.9) | 2/78 | (2.6) |
| *Family status* | | | | |
| never married | 70/297 | (23.6) | 21/74 | (28.4) |
| married | 155/297 | (52.2) | 34/74 | (45.9) |
| divorced | 45/297 | (15.2) | 13/74 | (17.6) |
| widowed | 27/297 | (9.1) | 6/74 | (8.1) |
| *Formal education* | | | | |
| low | 134/301 | (44.5) | 40/77 | (51.9) |
| medium | 103/301 | (34.2) | 27/77 | (35.1) |
| high | 64/301 | (21.3) | 10/77 | (13.0) |
| *Mother tongue* | | | | |
| German | 275/300 | (91.7) | 75/76 | (98.7) |
| other | 25/300 | (8.3) | 1/76 | (1.3) |
| *Occupation* | | | | |
| employed | 142/300 | (47.3) | 33/75 | (44.0) |
| retired | 111/300 | (37.0) | 28/75 | (37.3) |
| homemaker | 12/300 | (4.0) | 4/75 | (5.3) |
| student | 11/300 | (3.7) | 2/75 | (2.7) |
| unemployed | 22/300 | (7.3) | 8/75 | (10.7) |
| other | 2/300 | (0.7) | - | - |
| *Health problem consulted* | | | | |
| type 2 diabetes | 110/308 | (35.7) | 31/78 | (39.7) |
| chronic back pain | 104/308 | (33.8) | 23/78 | (29.5) |
| depressive disorder | 82/308 | (26.6) | 21/78 | (26.9) |
| other | 12/308 | (3.9) | 3/78 | (3.8) |

[a] valid sample size varies between 297 and 308 due to missing values

[b] valid sample size varies between 74 and 78 due to missing values

related to SDM competence ratings with the OPTION-12 and the OPTION-5 and its out-strength was significantly positively related to the SDM competence rating with the 4HCS (Table 3). This means that how often this skill was used and how strongly it was associated with other skills could predict observer-rated SDM competence. Further, the outstrength of 'deliberating the decision' was significantly negatively associated with SDM competence as

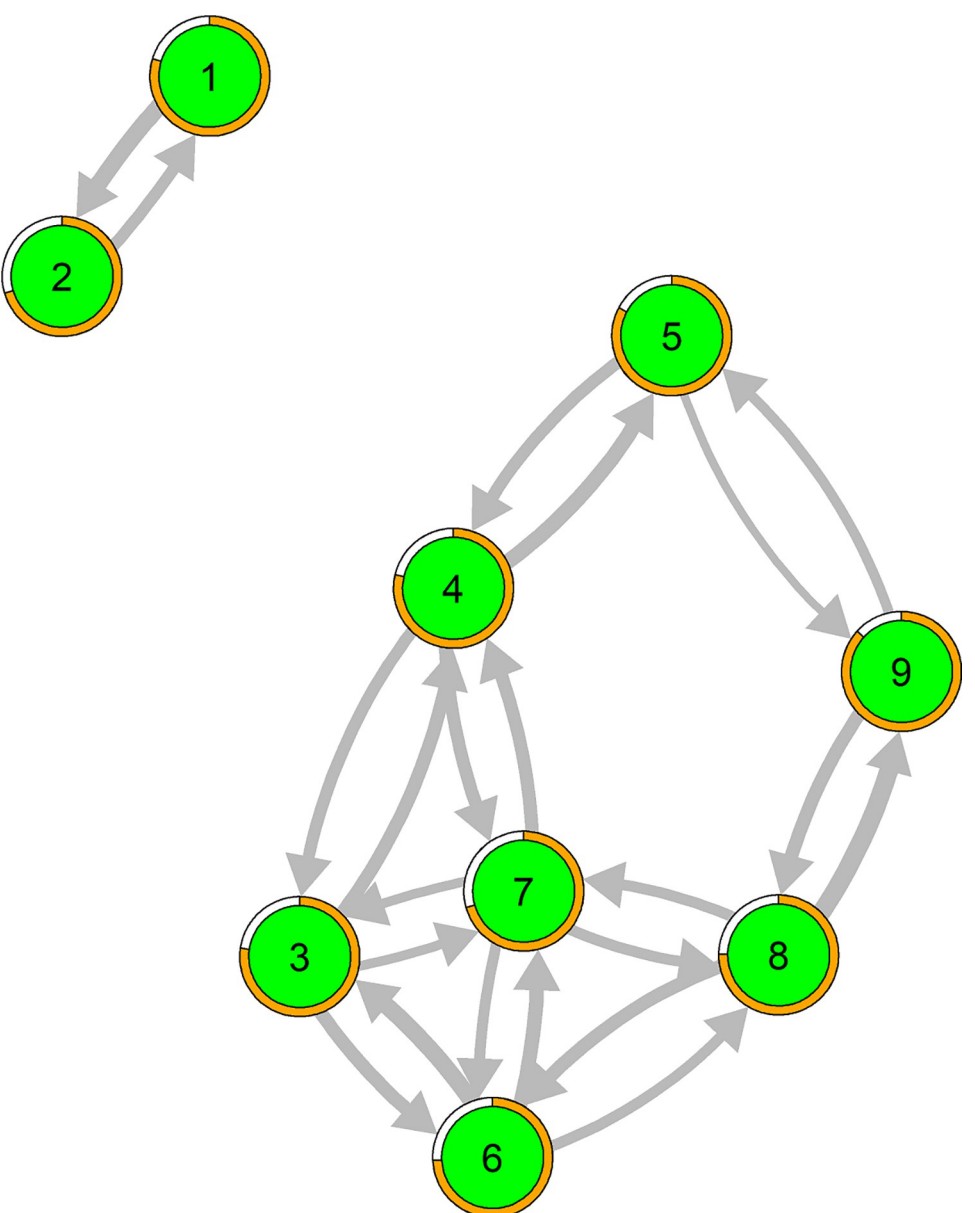

**Fig 2. Average skills network across physicians.** The width of the arrows represents the strength of the skills associations. The pie around each node indicates the extent of activation of each item. The labels refer to the following skills: 1. focusing the decision; 2. sharing the decision; 3. presenting options; 4. informing on options; 5. supporting comprehension; 6. eliciting preferences; 7. deliberating the decision; 8. selecting an option; 9. planning actions.

measured with the OPTION-12. This indicated that when a physician's network showed that 'deliberating the decision' influenced many other skills, the SDM competence of that physician was rated lower. The confirmatory model explained about half of the variance of the observer-rated SDM competence with correlations between predicted and observed values ranging from 0.65 to 0.75. Thus, skills network characteristics explained a considerable amount of variation in the observer-rated SDM competence of physicians. Predicted and observed values of the confirmatory models are depicted in Fig 4, panels A-C.

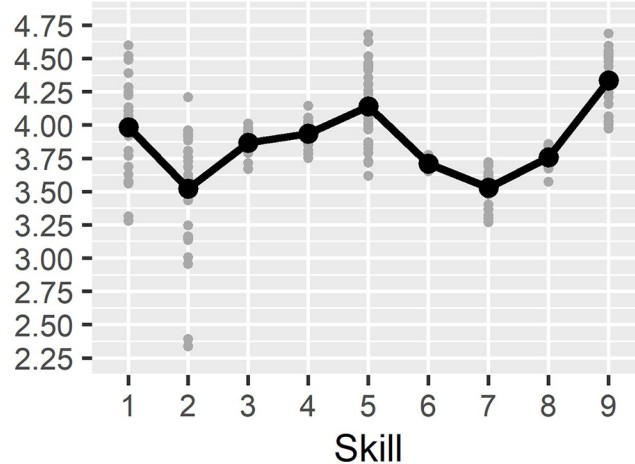

**A** Activation

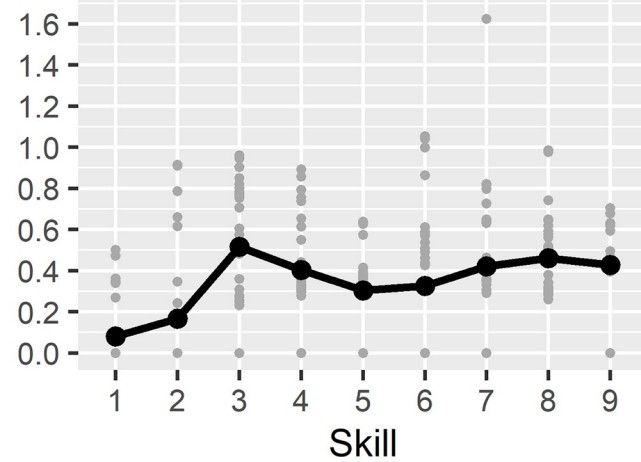

**B** Outstrength

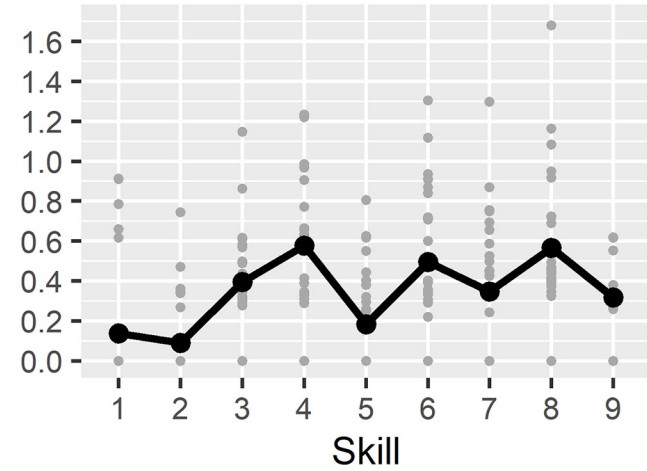

**C** Instrength

**Fig 3. Network parameters of the investigated skills.** Black dots represent the average score, and grey dots indicate estimates from each physician network. The labels refer to the following skills: 1. focusing the decision; 2. sharing the decision; 3. presenting options; 4. informing on options; 5. supporting comprehension; 6. eliciting preferences; 7. deliberating the decision; 8. selecting an option; 9. planning actions.

## Exploratory prediction of observed SDM competence from skills networks

When the activation, instrength and outstrength of all skills were regressed on the observer-rated SDM competence, the skills ´focusing on the decision´, 'presenting option', 'informing on options' and 'eliciting preferences' were significantly related to at least one of the three observer measures (S1–S3 Tables). Results from the subsequent analysis, which included the activation, instrength and outstrength of these four skills, are reported in Table 4. Only the activation of 'eliciting preference' was significantly related to SDM competence as measured by the OPTION-5. Still, the model explained about half of the variance for each of the observer measures, with multiple correlation coefficients ranging from 0.69 to 0.82 (Table 4). Predicted and observed values of the exploratory models are displayed in Fig 4, Panels D-F.

## Discussion

A wide range of empirical results suggest that physicians have a limited ability to assess their professional competences accurately [27]. This includes communication competences, where studies frequently show a lack of association between physicians' self-assessment and external rating by trained observers [28–30]. Here, we found encouraging evidence that it is possible to use physicians' self-assessment of behavioral skills for measuring competence, even though the measurement is computationally more complex than using simple (averaged) global ratings as a direct measure of competence.

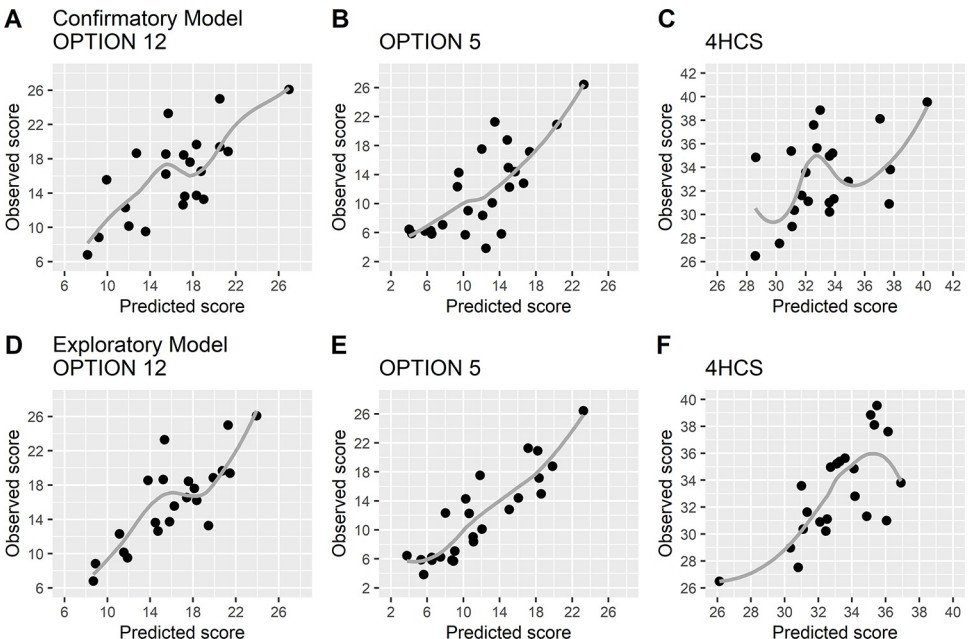

**Fig 4. Calibration plots for the confirmatory and exploratory prediction of observer-rated SDM competence.** Panels A, B and C show predicted and observed scores for the confirmatory model, panels D, E and F for the exploratory model. Black dots represent the physicians' scores; smoothing (loess) curves are displayed for each outcome by the grey line. 4HCS, Four Habits Coding Scheme.

**Table 3. Confirmatory prediction of observed SDM competence from network parameters.**

| | OPTION-12 (n = 22) | OPTION-5 (n = 24) | 4HCS (n = 22) |
|---|---|---|---|
| | Estimate [95% CI] | Estimate [95% CI] | Estimate [95% CI] |
| Intercept | 15.96 [14.67 to17.23] | 11.96 [10.31 to 13.51] | 33.07 [32.02 to 34.10] |
| *Activation* | | | |
| Skill 1 | 4.18 [-1.84 to 9.81] | 2.16 [-4.67 to 9.05] | 1.51 [-3.72 to 6.61] |
| Skill 6 | 43.44 [0.32 to 86.45]* | 65.62 [9.98 to 119.88]* | 18.84 [-14.33 to 51.00] |
| Skill 7 | 2.51 [-14.64 to 20.22] | 7.90 [-11.54 to 27.26] | 7.96 [-6.18 to 23.66] |
| *Instrength* | | | |
| Skill 1 | 2.91 [-5.92 to 11.71] | -3.47 [-13.35 to 6.75] | 4.46 [-2.52 to 11.04] |
| Skill 6 | 2.32 [-4.47 to 8.83] | 7.42 [-0.27 to 15.75] | 1.38 [-3.57 to 6.44] |
| Skill 7 | 1.43 [-4.37 to 7.10] | 1.89 [-4.98 to 9.07] | -1.69 [-6.73 to 2.77] |
| *Outstrength* | | | |
| Skill 1 | -7.15 [-16.63 to 2.64] | -5.78 [-19.25 to 7.83] | -5.38 [-12.18 to 1.20] |
| Skill 6 | 3.16 [-2.85 to 9.05] | 0.32 [-6.97 to 7.43] | 4.70 [0.12 to 9.70]* |
| Skill 7 | -6.30 [-11.57 to -1.00]* | -4.77 [-11.20 to 1.98] | -1.40 [-5.62 to 2.77] |
| *R* | 0.750 [0.620 to 0.812] | 0.750 [0.619 to 0.819] | 0.653 [0.504 to 0.732] |
| $R^2$ | 0.563 [0.385 to 0.660] | 0.562 [0.383 to 0.670] | 0.427 [0.254 to 0.536] |

Note: skill 1, focusing the decision; skill 6, eliciting preferences; skill 7, deliberating the decision; CI = credible interval; 4HCS, Four Habits Coding Scheme; R, multiple correlation, $R^2$, explained variance

* With a probability of at least 95%, this parameter is different from zero.

In the population network, the most central SDM skills were presenting options, informing on options, eliciting preferences, deliberating the decision, and selecting an option. Supporting comprehension and planning actions seem to be somewhat more peripheral skills, while focusing the decision and sharing the decision are (albeit strongly associated with each other) completely disconnected from the rest of the network. This architecture is strikingly similar to the structure of the population network of SDM skills based on patient-reported data [13], even though patient and physician assessments of the specific skills from the same consultation considerably disagreed in previous investigations [31, 32]. It should also be noted that, although we did not attempt to cluster skills in the present study explicitly, the identified structure of the SDM skills shows similarities with the categorization of the skills postulated by the three-talk model of SDM by Elwyn and colleagues [33]. These findings suggest that skills networks are able to capture a robust and replicable physician-level construct, which we hypothesize to be SDM competence.

Validity of interpreting the information contained in the network structure as an indicator of SDM competence was supported by its association with observer-rated data. In a confirmatory approach, we found that the combination of data-based inference with findings from the analysis of patient-reported data [13] (in the form of informative priors for Bayesian analysis) produced strong predictions of observer-rated competence. In the spirit of a continuous Bayesian accumulation of evidence, the results of the confirmatory analysis can be considered to synthesize the findings of the previously reported investigation using patient-reported data and the current study based on physicians' self-assessment quantitatively. Results of the exploratory analysis led to models with even stronger predictive accuracy. This indicates that skills networks based on physicians'self-assessment of their SDM skills were highly predictive of their SDM competence as rated by external observers. In general, the findings support the hypothesis that patient and physician rated data may be used interchangeably for competence assessment if handled in the context of the network approach.

**Table 4. Exploratory prediction of observed SDM competence from network parameters.**

| | OPTION-12 (n = 22) | OPTION-5 (n = 24) | 4HCS (n = 22) |
|---|---|---|---|
| | Estimate [95% CI] | Estimate [95% CI] | Estimate [95% CI] |
| Intercept | 15.84 [13.50 to 17.86] | 11.77 [9.84 to 13.68] | 33.07 [31.04–34.98] |
| *Activation* | | | |
| Skill 1 | 2.10 [-7.02 to 10.96] | 1.11 [-7.25 to 8.79] | 0.50 [-7.31–8.09] |
| Skill 3 | 12.48 [-27.76 to 52.85] | 22.21 [-11.11 to 52.92] | 1.42 [-33.00–34.40] |
| Skill 4 | 2.34 [-33.13 to 35.67] | 15.47 [-15.52 to 42.52] | -4.13 [-33.19–26.41] |
| Skill 6 | 18.76 [-89.19 to 121.36] | 100.09 [2.61 to 189.00]* | -17.78 [-108.27–72.45] |
| *Instrength* | | | |
| Skill 1 | 4.93 [-18.16 to 28.10] | -3.78 [-24.06 to 16.69] | 13.98 [-6.47–33.00] |
| Skill 3 | 5.39 [-8.39 to 19.41] | -2.69 [-12.15 to 7.10] | 4.52 [-6.90–16.28] |
| Skill 4 | -3.57 [-16.50 to 9.43] | -4.01 [-13.10 to 5.69] | -1.17 [-12.84–9.89] |
| Skill 6 | 1.59 [-8.31 to 11.51] | -1.65 [-9.80 to 7.12] | 3.75 [-4.71–12.66] |
| *Outstrength* | | | |
| Skill 1 | -19.27 [-61.96 to 23.77] | -8.06 [-47.55 to 30.42] | -28.64 [-65.74–11.31] |
| Skill 3 | -1.73 [-15.10 to 10.86] | 4.40 [-4.85 to 13.06] | -3.95 [-15.36–7.29] |
| Skill 4 | -3.09 [-15.39 to 9.14] | -4.37 [-13.61 to 4.85] | -2.38 [-13.53–8.33] |
| Skill 6 | 4.72 [-3.42 to 12.63] | 5.96 [-1.17 to 13.20] | 0.86 [-5.92–7.79] |
| *R* | 0.755 [0.620 to 0.829] | 0.821 [0.700 to 0.871] | 0.693 [0.559 to 0.775] |
| $R^2$ | 0.570 [0.384 to 0.686] | 0.674 [0.490 to 0.785] | 0.480 [0.312 to 0.600] |

Note: skill 1, focusing the decision; skill 3, presenting options; skill 4, informing on options; skill 6, eliciting preferences; CI, credible interval; 4HCS, Four Habits Coding Scheme; R, multiple correlation, $R^2$, explained variance

* With a probability of at least 95%, this parameter is different from zero.

Both patients'and physicians'ratings of SDM processed according to the skills network approach seem to yield an objective assessment of SDM competence, which highly relates to external assessments of this competence. This finding has various implications. From a theoretical perspective, it suggests a new definition of professional competence, which can be contrasted to and integrated with existing ones [34]. For the network science of psychological phenomena [35], it means a methodological extension and a new field of application. Lastly, for assessing professional SDM competence [36], it offers a new way of measurement based on self-rating of physicians. By applying the skills network model of SDM competence to physician-rated data, we provided a promising opportunity for a feasible assessment of SDM competence. Self-ratings are, in contrast to observer ratings, more easily applicable and less time-intensive, offering a genuine opportunity for their application in routine practice.

Since measuring SDM competence with skills networks seem to offer a replicable and robust assessment of this professional skill (high agreement between patient, physician, and observer assessment), our proposed method is of relevance and could be applied to areas in which a feasible and robust assessment of SDM competence is highly needed. First, research on SDM depends largely on a valid measurement of SDM competence, for example to assess predictors and treatment outcomes for different levels of SDM competence. Considering that assessing competence by observation is very resource intensive, utilizing brief self-assessment increases the range of options for research projects. Second, to evaluate the effectiveness of a trainings for SDM, including education of health care professionals, the assessment of this competence is of central importance. Novel measures without the need for external judgment by qualified experts could contribute to a more comprehensive evaluation of interventions

aiming to implement SDM. Finally, the network approach to SDM competence could be applied when assessing SDM as a part of quality management in clinical routine care. Here, a robust assessment can be gained from quite easily attainable patient or physician ratings of SDM. In this context, analysis of a continuous data stream from SDM surveys may enable monitoring of the SDM competence of individuals, teams, departments, or hospitals. Furthermore, a detailed analysis of the obtained skills networks could reveal specific and actionable targets (i.e., skills or skill connections) for improvement. Being able to create individual skills networks and to precisely pinpoint skills and skill connections that need to be improved could open the way to a data-driven and individualized measurement, education, training, and monitoring of complex competences.

Current findings are limited by the restricted sample size and the considerable complexity of the statistical models relative to the sample size. These factors are likely to be partly responsible for the wide credible intervals of the estimated parameters. Due to this imprecision and to collinearity between network parameters, the influence of specific network parameters of individual skills could be investigated only to a limited extend. As network parameters were correlated to each other, it remains unclear how each individual network parameter relates to observer-rated SDM competence and which network parameters are most important for indicating SDM competence. Jointly, the network parameters showed a high predictive accuracy for the observer-rated SDM competence, and future studies need to assess which specific network parameters are most important for this. Furthermore, since the approach has been only applied to data from a self-selected sample from outpatient care in Germany, generalizability to other contexts needs to be investigated in future studies. This should also include comparing results between various contexts and subgroups, for example, defined by the primary specialty of the physician or the disease of the consulted patients, which was unfortunately not possible in the present study due to the limited sample size. Finally, results from the exploratory analyses need to be interpreted with due caution, as different model building procedures could have led to different results and current results could not be cross-validated. Still, especially through the confirmatory testing and the replication of findings from previous analyses with patient data, the current study offered considerable support for the skills network approach to SDM competence. By applying a Bayesian framework, some previously mentioned weaknesses could be extenuated and problems such as the multiple testing problem avoided. Future studies need to test this new approach with larger datasets to assess the relative importance of individual network parameters and skills.

Structuring clinical competences into a hierarchically organized categorical system is challenging, particularly in the interpersonal and communication domains [37]. "Choosing the right boundaries for a unit of analysis is a central problem in every science" [38], and this is particularly true for clinical skills and competences, which are strongly interrelated and frequently overlapping. Thus, it is not always clear how to narrow down the densely connected network of clinical skills into well definable and analyzable competences. Whether SDM is a sufficiently distinct concept from this perspective, i.e., whether it is operationally sufficiently closed in the environment of other skills and competences, should be empirically investigated in further studies by collecting data on a broader range of skills and competences for network analysis.

## Conclusions

Our findings provide further support for conceptualizing and modeling physicians' SDM competence as a network of SDM skills. This conceptualization suggests a new definition of professional competence, offers a methodological extension and a new field of application for

network science and, most importantly, provides a new way of measuring professional competence based on self-rating of physicians. A robust measurement of SDM competence offers new opportunities for research, for evaluating learning success in education and training, and for monitoring SDM competence for quality management purposes in clinical routine care. In combination, these consistent theoretical, empirical, and practical implications have the potential to open up a new approach to professional competence in health care.

## Supporting information

**S1 File. Information on prior distributions.**
(PDF)

**S1 Fig. Skills networks of individual physicians.**
(PDF)

**S1 Table. Prediction of observer-rated shared decision making competence from the activation of all skills.**
(PDF)

**S2 Table. Prediction of observer-rated shared decision making competence from the outstrength of all skills.**
(PDF)

**S3 Table. Prediction of observer-rated shared decision making competence from the instrength of all skills.**
(PDF)

## Author Contributions

**Conceptualization:** Levente Kriston.

**Data curation:** Levente Kriston, Lea Schumacher, Isabelle Scholl.

**Formal analysis:** Levente Kriston, Lea Schumacher.

**Funding acquisition:** Levente Kriston, Martin Härter.

**Investigation:** Levente Kriston, Lea Schumacher, Pola Hahlweg, Martin Härter, Isabelle Scholl.

**Methodology:** Levente Kriston, Lea Schumacher.

**Project administration:** Levente Kriston, Isabelle Scholl.

**Resources:** Martin Härter.

**Supervision:** Levente Kriston, Isabelle Scholl.

**Validation:** Levente Kriston, Pola Hahlweg, Martin Härter, Isabelle Scholl.

**Visualization:** Levente Kriston, Lea Schumacher.

**Writing – original draft:** Levente Kriston, Lea Schumacher.

**Writing – review & editing:** Pola Hahlweg, Martin Härter, Isabelle Scholl.

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
