## [Decision Letter · Decision Letter 0]

7 Mar 2022

PONE-D-21-25125Application of the skills network approach to measure physician competence in shared decision making based on self-assessmentPLOS ONE

Dear Dr. Kriston

Thank you for submitting your manuscript to PLOS ONE. After careful consideration, we feel that it has merit but does not fully meet PLOS ONE’s publication criteria as it currently stands. Therefore, we invite you to submit a revised version of the manuscript that addresses the points raised during the review process.

Please submit your revised manuscript by April 21, 2022. If you will need more time than this to complete your revisions, please reply to this message or contact the journal office at plosone@plos.org. Please include the following items when submitting your revised manuscript:A rebuttal letter that responds to each point raised by the academic editor and reviewer(s). You should upload this letter as a separate file labeled 'Response to Reviewers'.A marked-up copy of your manuscript that highlights changes made to the original version. You should upload this as a separate file labeled 'Revised Manuscript with Track Changes'.An unmarked version of your revised paper without tracked changes. You should upload this as a separate file labeled 'Manuscript'.

We look forward to receiving your revised manuscript.

Kind regards,

Edris Hasanpoor

Academic Editor

PLOS ONE

Journal Requirements:

2. Thank you for stating the following in the Competing Interests section: "I have read the journal's policy and the authors of this manuscript have the following competing interests: LK, MH, and IS report academic, but not financial, conflict of interest as the developers of the investigated measure, the SDM-Q-Doc."

Reviewers' comments:

Reviewer's Responses to Questions

**Comments to the Author**

**Reviewers’ comments**

he article focuses on the assessment whether using the skills network approach allows for predicting observer-rated shared decision making (SDM) competence of physicians from their self-reported SDM skills. The article is well written, methodologically well done, and focuses on an important topic. However, I would like to suggest some points that should be added or slightly changed.

Abstract - Conclusion: You conclude that SDM skill ratings offer new theoretically and empirically grounded opportunities for the assessment of SDM competence in routine medical care. Please add practical implications.

Introduction:

The introduction is written very well. I only suggest to add some sentences with regard to the worth of shared decision making (SDM) in medical consultations.

Methods:

I suggest to display your study design and measures graphically. This would give the reader a good overview on the design and the different measures.

Results

The presentation of results is well done and well implemented in tabular and graphical form. No further suggestions.

Discussion:

From my point of view the discussion section should be widened. I suggest to add some practical implications that can be drawn from your results.

Please discuss collinearity between network parameters in more detail.

Conclusion

From my point of view, I suggest leaving out the references in the summary. They should rather appear in the discussion and support it.

Data Availability: Please give some more explanations for the restrictions that apply.

I am wondering why your analyses are based on data collected in 2009/2010. Could you shortly explain the time delay.

---

## [Author Response · Author response to Decision Letter 0]

6 Apr 2022

# Reviewer 1

1. “The article focuses on the assessment whether using the skills network approach allows for predicting observer-rated shared decision making (SDM) competence of physicians from their self-reported SDM skills. The article is well written, methodologically well done, and focuses on an important topic. However, I would like to suggest some points that should be added or slightly changed.”

- Thank you very much for the overall positive assessment of our study. 

2. “Abstract - Conclusion: You conclude that SDM skill ratings offer new theoretically and empirically grounded opportunities for the assessment of SDM competence in routine medical care. Please add practical implications.”

- Thank you for this comment. We added information on the practical implications of the skills network approach for SDM in the Abstract, p.3. 

3. “Introduction: The introduction is written very well. I only suggest to add some sentences with regard to the worth of shared decision making (SDM) in medical consultations.”

- We added a paragraph at the beginning of the Introduction on the value of SDM describing its ethical imperative and possible benefits, p.4.

4. “Methods: I suggest to display your study design and measures graphically. This would give the reader a good overview on the design and the different measures.”

- Thank you very much for this idea. We created a figure on the design, measures and analysis which we added in the Methods section, p.6 Figure 1. 

5. “Results: The presentation of results is well done and well implemented in tabular and graphical form. No further suggestions.”

- Thank you for this positive feedback. 

6. “Discussion: From my point of view the discussion section should be widened. I suggest to add some practical implications that can be drawn from your results.”

- We extended the Discussion section in regards to theoretical and practical implications of the skills network approach to SDM, p.15. 

7. “Please discuss collinearity between network parameters in more detail.

- Within the Discussion section, we further specified the limitations due to collinearity and the need for more research on individual network parameters, p.16. 

8. “Conclusion: From my point of view, I suggest leaving out the references in the summary. They should rather appear in the discussion and support it.”

- Thank you for this comment. We incorporated the references within the Discussion section and rewrote the conclusion without references, p. 17. 

9. “Data Availability: Please give some more explanations for the restrictions that apply.”

- The restrictions are necessary due to the limited resources that are (and expected to be) available for the workload associated with data sharing.

10. “I am wondering why your analyses are based on data collected in 2009/2010. Could you shortly explain the time delay.”

- As this study is a secondary data analysis, the original data collection (and the main analysis) of the data is quite long ago. Additionally, the skills network approach to SDM was only recently developed (Kriston L, Hahlweg P, Härter M, Scholl I. A skills network approach to physicians’ competence in shared decision making. Health Expect. 2020;23: 1466–1476. doi:10.1111/hex.13130). Therefore, we were interested in reanalyzing this data using this new approach.

---

## [Decision Letter · Decision Letter 1]

30 May 2022

PONE-D-21-25125R1Application of the skills network approach to measure physician competence in shared decision making based on self-assessmentPLOS ONE

Dear Dr. Kriston

Thank you for submitting your manuscript to PLOS ONE. After careful consideration, we feel that it has merit but does not fully meet PLOS ONE’s publication criteria as it currently stands. Therefore, we invite you to submit a revised version of the manuscript that addresses the points raised during the review process. Please submit your revised manuscript by 30, June. If you will need more time than this to complete your revisions, please reply to this message or contact the journal office at plosone@plos.org. Please include the following items when submitting your revised manuscript:A rebuttal letter that responds to each point raised by the academic editor and reviewer(s). You should upload this letter as a separate file labeled 'Response to Reviewers'.A marked-up copy of your manuscript that highlights changes made to the original version. You should upload this as a separate file labeled 'Revised Manuscript with Track Changes'.An unmarked version of your revised paper without tracked changes. You should upload this as a separate file labeled 'Manuscript'.If applicable, we recommend that you deposit your laboratory protocols in protocols.io to enhance the reproducibility of your results. Protocols.io assigns your protocol its own identifier (DOI) so that it can be cited independently in the future. For instructions see: https://journals.plos.org/plosone/s/submission-guidelines#loc-laboratory-protocols. Additionally, PLOS ONE offers an option for publishing peer-reviewed Lab Protocol articles, which describe protocols hosted on protocols.io. Read more information on sharing protocols at https://plos.org/protocols?utm_medium=editorial-email&utm_source=authorletters&utm_campaign=protocols.

We look forward to receiving your revised manuscript.

Kind regards,

Edris Hasanpoor

Academic Editor

PLOS ONE

Reviewers' comments:

Reviewer's Responses to Questions

**Comments to the Author**

Originally, this research was a quite interesting article. However, I note several points that can be improved

1 Based on their previously published method, they aimed to to measure physician competence in shared decision making based on self-assessment. However, in my opinion, they should put some practical or real-world application based on their newly claimed method especially in abstract, discussion, and conclusion. This article was only provide many theoretically methods without their application in the real-world setting.

2 The language used in the article perhaps will be difficult to be followed by most readers since it is full of technical words without explanation the function and meaning. I suggest they can revise the language used

---

## [Author Response · Author response to Decision Letter 1]

20 Jun 2022

#1 Based on their previously published method, they aimed to to measure physician competence in shared decision making based on self-assessment. However, in my opinion, they should put some practical or real-world application based on their newly claimed method especially in abstract, discussion, and conclusion. This article was only provide many theoretically methods without their application in the real-world setting.

Response: Thank you very much for this suggestion. We have added a substantial amount of information on implications for practical and real-world applications to the revised manuscript (Abstract, Discussion, Conclusions).

#2 The language used in the article perhaps will be difficult to be followed by most readers since it is full of technical words without explanation the function and meaning. I suggest they can revise the language used.

Response: Thank you for keeping the readers’ technical skills in mind. We have added non-technical explanations to several points and revised the language substantially throughout the manuscript to improve comprehensibility.

---

## [Decision Letter · Decision Letter 2]

12 Dec 2022

PONE-D-21-25125R2Application of the skills network approach to measure physician competence in shared decision making based on self-assessmentPLOS ONE

Dear Dr. Kriston,

Thank you for submitting your manuscript to PLOS ONE. After careful consideration, we feel that it has merit but does not fully meet PLOS ONE’s publication criteria as it currently stands. Therefore, we invite you to submit a revised version of the manuscript that addresses the points raised during the review process.

I am aware that you have waited a long time for this decision, and I apologise for this. Unfortunately, the original Academic Editor and reviewers became unavailable, apart from reviewer 1, who has reassessed the manuscript and is very happy with the result, but asks only that all data is shared, as per PLOS ONE data sharing policies. As such, two new reviewers were invited, who only have minor suggestions to improve the strength of the manuscript. 

Reviewer 5 provides detailed comments, which we ask that you consider carefully.

Reviewer 4 suggests that the manuscript might be difficult for a layperson to utilise. This limitation would not preclude consideration for publication in PLOS ONE, and I leave this with you to assess whether you prefer to address this in the revised manuscript, or whether you choose to otherwise make the findings from your study accessible to a layperson audience or clinicians interested in the topic, such as through a preprint, on your own website, Figshare etc. You may also add this information as a "Comment" on the final paper at a later stage, should it be published in PLOS ONE.

We look forward to receiving your revised manuscript.

Kind regards,

Hanna Landenmark

Staff Editor

PLOS ONE

Journal Requirements:

Reviewers' comments:

Reviewer's Responses to Questions

**Comments to the Author**

1. If the authors have adequately addressed your comments raised in a previous round of review and you feel that this manuscript is now acceptable for publication, you may indicate that here to bypass the “Comments to the Author” section, enter your conflict of interest statement in the “Confidential to Editor” section, and submit your "Accept" recommendation.

Reviewer #1: All comments have been addressed

Reviewer #4: (No Response)

Reviewer #5: (No Response)

2. Is the manuscript technically sound, and do the data support the conclusions?

Reviewer #1: Yes

Reviewer #4: Partly

Reviewer #5: Yes

3. Has the statistical analysis been performed appropriately and rigorously? 

Reviewer #1: Yes

Reviewer #4: I Don't Know

Reviewer #5: Yes

4. Have the authors made all data underlying the findings in their manuscript fully available?

Reviewer #1: No

Reviewer #4: No

Reviewer #5: (No Response)

5. Is the manuscript presented in an intelligible fashion and written in standard English?

Reviewer #1: Yes

Reviewer #4: Yes

Reviewer #5: Yes

6. Review Comments to the Author

Reviewer #1: All comments have been addressed. Perhaps, complete data can be shared using data sharing platform such as figshare, etc

Reviewer #4: The authors indicate that they have added real world practical value, but it's not easy to glean. There remains a lot of technical and statistical jargon that would be difficult for a layperson to access.

Reviewer #5: The paper has very interesting data and a well-organized analysis. The goal is to improve the SDM definition by understanding how physicians' behavioral skills could be organized into patterns to help patients make medical decisions.

Measures

Their description of qualitative analysis is only briefly described. For instance, could you please elaborate on OPTION-5 and OPTION-12 measures? OPTION-5 is a shorter version of OPTION-12 what was the rationale to use both measures? What are the examples of the items?

Results

Could you please report the qualitative results of transcript coding using the standard guidelines (O’Brien et al., 2014)? What is interrater reliability? What were the key examples of observed competencies?

O’Brien, Bridget C. PhD; Harris, Ilene B. PhD; Beckman, Thomas J. MD; Reed, Darcy A. MD, MPH; Cook, David A. MD, MHPE. Standards for Reporting Qualitative Research: A Synthesis of Recommendations. Academic Medicine: September 2014 - Volume 89 - Issue 9 - p 1245-1251 doi: 10.1097/ACM.0000000000000388

SDM Q9 and Option 5; Option 12; Are created based on Ewyn’s model of SDM. How do the clusters identified in network analysis speak to this model?

How stable are discovered patterns in the network? If physician rating is regressed only on transcripts data of patients with diabetes, or only with depression, do you see the same patterns?

Discussion

Network analysis shows that relationships between skills need to be considered to ensure that skills could predict observable competencies. To what extent do specific patterns matter in this analysis versus the presence or absence of skills? How could the discovery of patterns in the physicians’ skills contribute to physicians’ education and practice?

7. PLOS authors have the option to publish the peer review history of their article (what does this mean?). If published, this will include your full peer review and any attached files.

Reviewer #1: **Yes: **Edwin Njoto

Reviewer #4: No

Reviewer #5: No

---

## [Author Response · Author response to Decision Letter 2]

12 Jan 2023

EDITORIAL COMMENTS

E#1. Thank you for submitting your manuscript to PLOS ONE. After careful consideration, we feel that it has merit but does not fully meet PLOS ONE’s publication criteria as it currently stands. Therefore, we invite you to submit a revised version of the manuscript that addresses the points raised during the review process. I am aware that you have waited a long time for this decision, and I apologise for this. Unfortunately, the original Academic Editor and reviewers became unavailable, apart from reviewer 1, who has reassessed the manuscript and is very happy with the result, but asks only that all data is shared, as per PLOS ONE data sharing policies. As such, two new reviewers were invited, who only have minor suggestions to improve the strength of the manuscript. 

Response to E#1. Thank you for taking this quite long process in your hands. We will share the data as recommended by Reviewer 1 and have implemented the suggestions of the two new reviewers.

E#2. Reviewer 5 provides detailed comments, which we ask that you consider carefully.

Response to E#2. We have revised the manuscript according to the suggestions of Reviewer 5.

E#3. Reviewer 4 suggests that the manuscript might be difficult for a layperson to utilise. This limitation would not preclude consideration for publication in PLOS ONE, and I leave this with you to assess whether you prefer to address this in the revised manuscript, or whether you choose to otherwise make the findings from your study accessible to a layperson audience or clinicians interested in the topic, such as through a preprint, on your own website, Figshare etc. You may also add this information as a "Comment" on the final paper at a later stage, should it be published in PLOS ONE.

Response to E#3. We have substantially improved readability for laypersons in two peer review rounds and are very happy with the result of this process. However, a further tailoring of the contents for laypersons would endanger the accuracy of the technical language. Therefore, we have decided to create a plain language summary of the study and make it publicly accessible. We have included the link to this summary in the revised article.

REVIEWER 1

R1#1. All comments have been addressed. Perhaps, complete data can be shared using data sharing platform such as figshare, etc.

Response to R1#1. Thank you for reviewing the revised manuscript. We will share the data and the analysis code of the study.

 

REVIEWER 4

R4#1. The authors indicate that they have added real world practical value, but it's not easy to glean. There remains a lot of technical and statistical jargon that would be difficult for a layperson to access.

Response to R4#1. Thank you for reviewing the manuscript. We feel that a further tailoring of the contents for laypersons would endanger the accuracy of the technical language. Therefore, we have decided to create a plain language summary of the study and make it publicly accessible. We have included the link to this summary in the revised article.

REVIEWER 5

R5#1. The paper has very interesting data and a well-organized analysis. The goal is to improve the SDM definition by understanding how physicians' behavioral skills could be organized into patterns to help patients make medical decisions.

Response to R5#1. Thank you for this kind feedback and the helpful comments on the manuscript.

R5#2. Measures: Their description of qualitative analysis is only briefly described. For instance, could you please elaborate on OPTION-5 and OPTION-12 measures? OPTION-5 is a shorter version of OPTION-12 what was the rationale to use both measures? What are the examples of the items?

Response to R5#2. The study did not include qualitative analyses. Nevertheless, we have added information on the measures to the Methods section of the revised manuscript, including the rationale for using three measures.

R5#3. Results: Could you please report the qualitative results of transcript coding using the standard guidelines (O’Brien et al., 2014)? What is interrater reliability? What were the key examples of observed competencies?

O’Brien, Bridget C. PhD; Harris, Ilene B. PhD; Beckman, Thomas J. MD; Reed, Darcy A. MD, MPH; Cook, David A. MD, MHPE. Standards for Reporting Qualitative Research: A Synthesis of Recommendations. Academic Medicine: September 2014 - Volume 89 - Issue 9 - p 1245-1251 doi: 10.1097/ACM.0000000000000388

Response to R5#3. The study did not include qualitative analyses, but we have added findings on the interrater reliability of the measures.

R5#4. SDM Q9 and Option 5; Option 12; Are created based on Ewyn’s model of SDM. How do the clusters identified in network analysis speak to this model?

Response to R5#4. According to our knowledge, Elwyn and colleagues have not specified an explicit model of how certain skills interact, although they did group them into loosely defined categories (‘team talk”, “option talks”, “decision talk”). In our study, we did not attempt to identify skill clusters. We have included this interesting line of though in the Discussion of the revised manuscript. 

R5#5. How stable are discovered patterns in the network? If physician rating is regressed only on transcripts data of patients with diabetes, or only with depression, do you see the same patterns?

Response to R5#5. This is a very important question. Unfortunately, the amount of data that were available for analysis is insufficient for performing subgroup analyses. Nevertheless, we added to the Discussion of the revised manuscript that possible moderators of the findings should be explored further.

R5#6. Discussion: Network analysis shows that relationships between skills need to be considered to ensure that skills could predict observable competencies. To what extent do specific patterns matter in this analysis versus the presence or absence of skills? How could the discovery of patterns in the physicians’ skills contribute to physicians’ education and practice?

Response to R5#6. If our results are replicated and prove to be robust in independent studies, this might be one of the most interesting implications of our findings. It would be possible to create a skills network for an individual physician or student (based on at least ten, better more, consultations) and pinpoint precisely the skill or the connection between skills that needs to be improved. This could lead to a data-driven individualized training and monitoring competences, showing the way towards a kind of “precision medical education”. We have included this thought in the Discussion of the revised manuscript.

---

## [Editor Report · Decision Letter 3]

14 Feb 2023

Application of the skills network approach to measure physician competence in shared decision making based on self-assessment

PONE-D-21-25125R3

Dear Dr. Kriston,

We’re pleased to inform you that your manuscript has been judged scientifically suitable for publication and will be formally accepted for publication once it meets all outstanding technical requirements.

Kind regards,

Hanna Landenmark

Staff Editor

PLOS ONE
---

## [Editor Report · Acceptance letter]

20 Feb 2023

PONE-D-21-25125R3 

Application of the skills network approach to measure physician competence in shared decision making based on self-assessment 

Dear Dr. Kriston:

I'm pleased to inform you that your manuscript has been deemed suitable for publication in PLOS ONE. Congratulations! Your manuscript is now with our production department. 

Kind regards, 

on behalf of

Dr. Edris Hasanpoor 

Academic Editor

PLOS ONE